# Offshore wind farm optimisation: a comparison of performance between regular and irregular wind turbine layouts

Maaike Sickler[1], Bart Ummels[1,2], Michiel Zaaijer[3], Roland Schmehl[3], and Katherine Dykes[4]

[1]Ventolines B.V., P.J. Oudweg 4, 1314 CH Almere, the Netherlands
[2]Faculty of Civil Engineering and Geosciences, Delft University of Technology, Stevinweg 1, 2628 CN Delft, the Netherlands
[3]Faculty of Aerospace Engineering, Delft University of Technology, Kluyverweg 1, 2629 HS Delft, the Netherlands
[4]DTU Wind Energy, Frederiksborgvej 399, 4000 Roskilde, Denmark

**Correspondence:** Roland Schmehl (r.schmehl@tudelft.nl)

**Abstract.** Layout optimisation is essential for improving the overall performance of offshore wind farms. During the past 15 years, the use of yield optimisation algorithms has resulted in a transition from regular to more irregular farm layouts. However, since the layout affects many factors, yield optimisation alone may not maximise the overall performance. In this paper, a comparative case study is presented to quantify the effect of the wind farm layout on the overall performance of offshore wind farms. The case study was performed to investigate two performance indicators: power performance, using yield calculations with WindPRO and wake-induced tower fatigue, using the Frandsen model. It is observed that irregular wind farm layouts have a higher annual energy production compared to regular layouts. Their power production is also more persistent (less sensitive) to wind direction, improving predictability and thus market value of power output. However, one turbine location in the irregular layout has a 24% higher effective turbulence level, leading to additional tower fatigue. As a result, fatigue-driven tower designs would require increased wall thicknesses, which would result in higher capital costs for all turbine locations. It is demonstrated in this study that layout optimisation using a minimum inter-turbine spacing effectively resolves the induced wake issue while maintaining high-yield performance.

## 1 Introduction

The share of wind energy in the electricity market is rapidly increasing (Musgrove, 2009; International Energy Agency (IEA), 2022). Offshore wind farms pose fewer geographical and social constraints than onshore wind farms, which leads to larger design spaces. The performance of an offshore wind farm indicates how efficient the system is at achieving its main objective (Tao and Finenko, 2016). Examining operational wind farms, a development of farm layouts over time can be recognised. Earlier wind farms show regular patterns such as the wind farms Horns Rev 1 (2002) (Akay et al., 2014) and Prinses Amalia (2008) (Stanley and Ning, 2019). Newer and larger wind farms show more variation in patterns such as the wind farms Horns Rev 2 (2009) (Ostachowicz et al., 2016) and Rødsand (2010) (Nygaard, 2014), and partial irregularity such as Anholt (2013) (Ostachowicz et al., 2016) and many more. A number of desktop optimisation studies even suggest fully irregular wind farms such as Research Layout 1 and 2 (Charhouni et al., 2019; Karouani and Elhoussaine, 2018) which are obtained from existing research as shown in Figure 1.

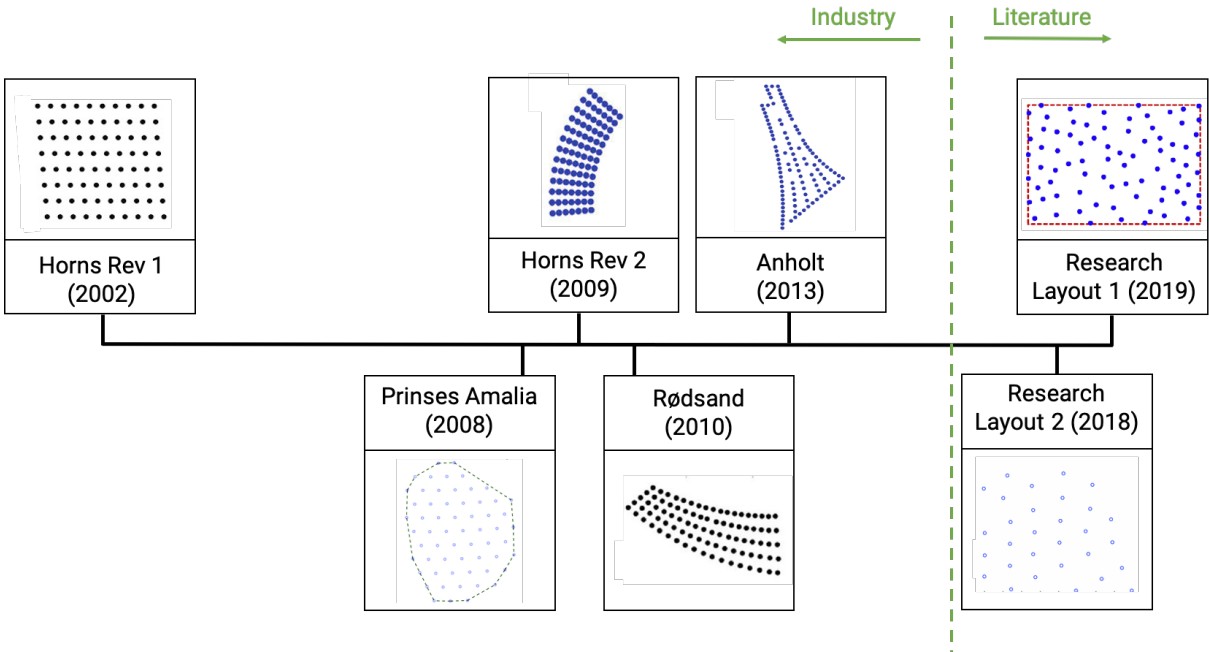

**Figure 1.** Development of wind farm patterns over time. Left of the green dashed line are wind farm layouts operational in industry (Akay et al., 2014; Stanley and Ning, 2019; Ostachowicz et al., 2016; Nygaard, 2014), while right of this line are optimised layouts from literature studies (Karouani and Elhoussaine, 2018; Charhouni et al., 2019).

Sanchez Perez Moreno (2019) investigated the preliminary design of the layout, electrical collection system, and support structures of an offshore wind farm using two different optimisation approaches. The sequential approach neglects the interaction between the three selected performance characteristics, while precisely this is taken into account in the multidisciplinary design analysis and optimisation (MDAO) approach. The two approaches were used to optimise the total system levelised cost of energy (LCOE) of a regular and an irregular farm layout. The study focussed on the comparison of the two approaches and the interaction effects, not comparing the performance of the different geometric patterns. Chen et al. (2015) used a multi-objective genetic algorithm (GA) to maximise the wind farm efficiency and minimise its cost applying real wind conditions. Investigating one regular and three irregular layouts with identical total geographical area, the comparison suggested that irregular geometric patterns may perform better than regular layouts, yet no final conclusion was drawn in the study. The goal of maximising the energy extraction while minimising the cost was also pursued by Charhouni et al. (2019) comparing regular and irregular wind farm layouts. The resulting power, capacity factor and efficiency were all higher for the irregular layout, although it should be noted that a constant wind speed and direction was considered. For general validity of this conclusion, the different layout options need to be investigated also at variable wind speed and direction.

Three observations can be made for comparisons of wind farms with regular and irregular layouts conducted to date. First, the performance indicators for wind farms are not well-defined in the literature. An overview of all possible indicators and how

they affect the overall performance is lacking. Also, the degree to which these indicators are influenced by the geometry of the wind farm is not investigated.

Second, the effects of optimised wind farm layouts on all performance indicators are either unknown or only partially investigated in the literature. The existing studies which include regular and irregular wind farm layouts focus on the performance of the optimisation tools. The aim of these optimisation studies is not to compare the overall performance of the regular and irregular wind farms.

Third, an optimisation of an existing regular wind farm pattern inherently leads to an increase in irregularity, as shown by optimisation studies (Grady et al., 2005; Marmidis et al., 2008; DuPont et al., 2012; Shakoor et al., 2016). This is only logical given the enormous design space of irregular patterns with many local optima compared to regular patterns. A particle swarm optimisation (PSO) or genetic algorithm (GA) is, for example, unlikely to find a regular pattern. Based on the nature of the optimisation algorithms many studies are naturally biased toward irregular wind farm layouts.

The objective of this paper is thus to quantify the effect of regular and irregular offshore wind farm layouts on selected performance indicators by means of a comparative case study using state-of-the-art simulation tools and models. The paper is based on the graduation project of the first author (Sickler, 2020) and is structured as follows. In Section 2Selection of performance indicatorssection.2, the selection of performance indicators is described. In Section 3Performance indicator group (1): power performancesection.3, the power production of the different layouts is assessed and in Section 4Performance indicator group (2): wake-induced tower fatiguesection.4 the wake-induced tower fatigue of the different layouts. In Section 5General applicability of resultssection.5, the general applicability of the results is investigated with a sensitivity analysis of the performance indicators for the Borssele wind farm. Conclusions are presented in Section 6Conclusionssection.6.

## 2 Selection of performance indicators

All performance indicators by which a wind farm has been assessed so far were inventorised using, among others, the works of Gonzalez et al. (2017) and Shafiee et al. (2016). The indicators were divided into three levels with the third level containing sub-categories, as shown in Figure 2. To incorporate changes in energy price, instead of using the LCOE as an overarching key performance indicator, profit (positive net present value) was selected (Nissen and Harfst, 2019).

To direct the research, a multi-criteria decision analysis was performed based on these criteria: (1) affected by wind farm layout, (2) feasibility to research, (3) site independence, and (4) technical nature. This resulted in the selection of five sub-performance indicators, which were grouped as to represent 'Power Performance'(yield/wake losses, predictability and value on the electricity market) and 'Wake-Induced Tower Fatigue (wind turbine cost, component replacement cost).

To assess the performance of these indicator groups for the different layout categories, the regular and irregular farm layouts depicted in Figure 3 were selected from the work of Sanchez Perez Moreno (2019). For both layouts, the IEA Wind Task 37 reference wind turbine with 10 MW rated power and a rotor diameter of 190.8 m was used (Bortolotti et al., 2019). The degree of irregularity was quantified mathematically with (1) the sum of the distance to surrounding turbines in a radius of 10 rotor diameters (10 RD) for each turbine and (2) the minimum inter-turbine spacing. The number of turbines with the same unique

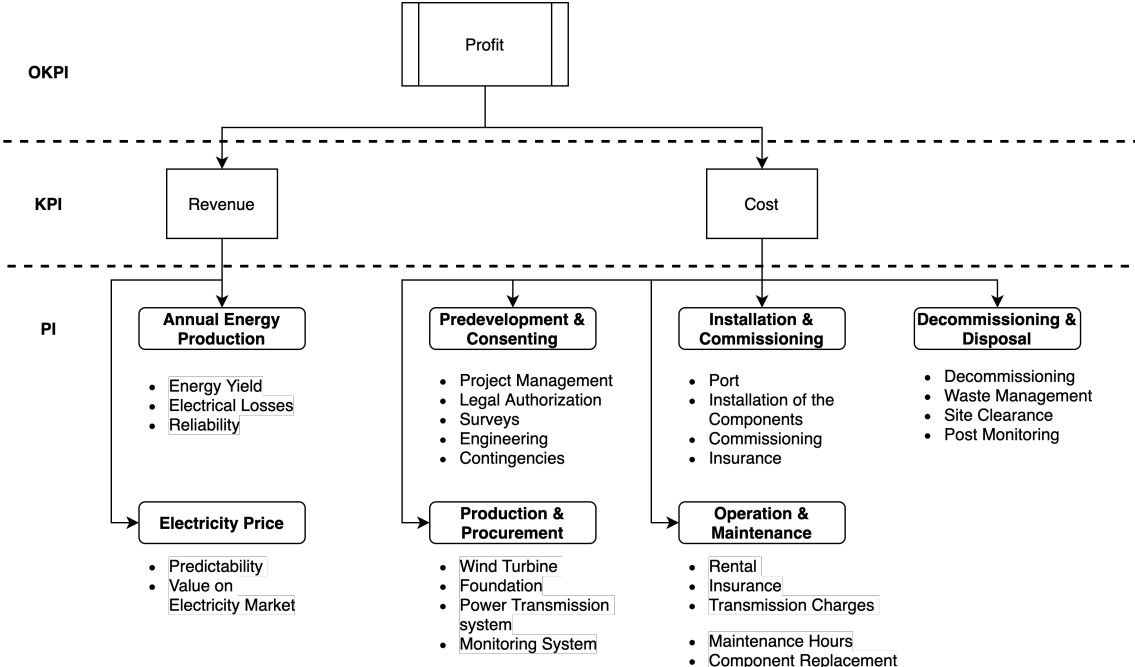

**Figure 2.** Breakdown of performance indicators. OKPI = overarching key performance indicator, KPI = key performance indicator, PI = performance indicator with itemised sub-performance indicators (Sickler, 2020).

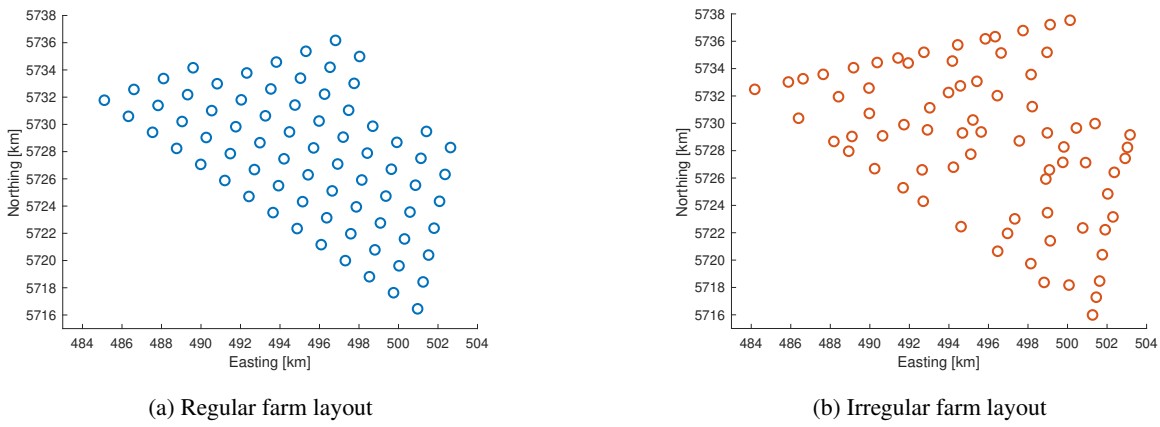

(a) Regular farm layout

(b) Irregular farm layout

**Figure 3.** Regular and irregular wind farm layouts from Sanchez Perez Moreno (2019) consisting of 74 turbines with a rated power of 10 MW at Borssele.

sum of distances to their surrounding turbines with a tolerance of 0.001 RD was 66 for the regular wind farm layout and only

5 for the irregular farm layout. The minimum inter-turbine spacing was 2.73 RD for the irregular wind farm layout compared to 8.88 RD for the regular wind farm.

The following two sections will present the analyses of performance indicator groups (1) and (2), respectively.

## 3 Performance indicator group (1): power performance

The annual energy production (AEP) was analysed in WindPRO, computing the absolute difference between regular and irregular wind farm layouts. The wind climate imported to WindPRO was obtained from Riezebos et al. (2015). The data was extrapolated to hub height using the power-law wind profile with a power exponent $\alpha$ of 0.08. The WindPRO calculation shows

a higher AEP of approximately 0.66% for the irregular wind farm layout, corresponding to approximately €700 000 according to the average European Power Exchange (EPEX) price in the Netherlands between 2007 and 2020. The individual turbine performance shows that the difference in AEP is not caused by the outliers (best-performing and worst-performing turbines) but by the average-performing turbines in the wind farm. Relating the performance and positions shows that the distribution of the lower performing turbines is more evenly spread for the regular wind farm than for the irregular wind farm as becomes

apparent from Figure 4.

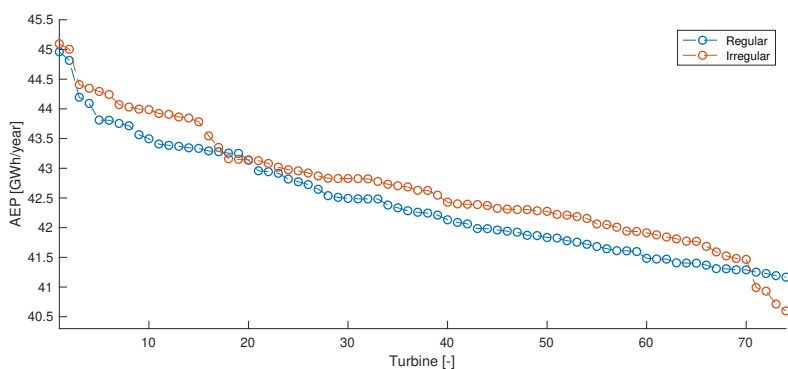

**Figure 4.** Individual turbine AEP for the regular and irregular wind farm layouts arranged from high performance to low performance (Sickler, 2020).

Figure 5 shows the persistence of power to wind direction, i.e. the extent to which power production varies with wind direction, measured in degrees, for a certain wind speed. The maximum power drop is quantified as the maximum uninterrupted decrease of power for an increase or decrease in wind direction. This maximum power drop decreases by 73.7 % for the irregular wind farm compared to the regular wind farm layout. The orientation of turbine rows in the wind farm with regular layout is

90 driving for the angle at which the power drops occur as well as for their magnitude. More turbines in a row correspond to a larger power drop. It is expected that a wind farm layout with a higher persistence to wind direction will lead to a decrease in prediction errors. This will likely result in lower imbalance costs. A rough estimation based on historical imbalance cost

data shows that the imbalance cost would amount to approximately 1.6 % of the AEP revenue. The difference between the imbalance cost of the regular and irregular wind farm would then become visible within this 1.6 %.

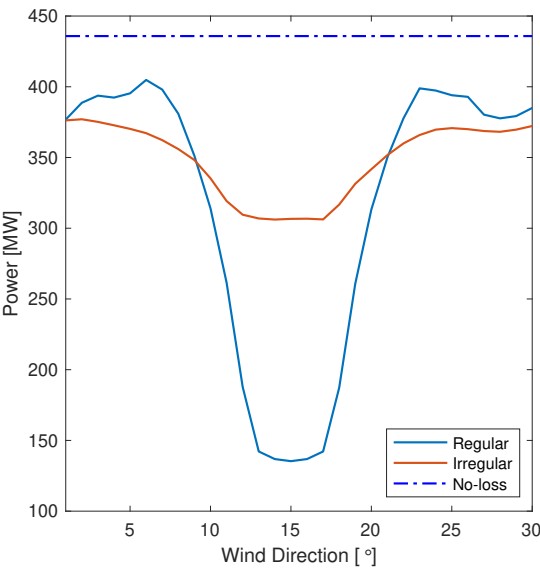

**Figure 5.** Power output of wind farms with regular and irregular layouts at a wind speed of 9.5 m/s as functions of wind direction, zoomed in on the wind directions with the largest power drops (Sickler, 2020).

The analysis for persistence to wind direction was executed using a constant mean wind speed of 9.5 m/s. This simplification likely overestimates the power drops as a function of wind direction because the time-dependent change of wake losses in the wind farm was not included. Additionally, the wind speed is just below the rated power, which means that the wake losses play a significant role. For higher wind speeds (15 m/s and above) the effect of wake losses is reduced or disappears.

    Based on the analysis and assumptions in this section, the irregular wind farm layout performs better for all three sub-
100 performance indicators analysed: the energy yield, predictability, and value in the electricity market. The importance and value of persistence to wind direction will increase as the impacts of wind power on power system operation increase with the very large growth foreseen in the next decades. The indicator predictability and value in the electricity market do require a much more extensive analysis for proper quantification.

## 4   Performance indicator group (2): wake-induced tower fatigue

The layout of a wind farm and the wind environment determine to what degree downstream turbines are affected by the wakes of upstream turbines. Especially for offshore farms, these wake effects are driving for the wind turbine fatigue loading (Thomsen and Sørensen, 1999). With low terrain roughness and low ambient turbulence intensities, the effect of wakes is higher than in onshore wind farms. Multiple studies confirm that one of the fundamental parameters which determine the wall thickness of

the tower design is fatigue (Igwemezie et al., 2018; Frandsen, 2007; Thomsen and Sørensen, 1999; Frohboese and Schmuck, 2010). Therefore, the layout of the farm will affect the cost of the towers.

Interestingly, while monopile foundation designs are optimised for individual locations within an offshore wind farm, typically only a single tower design is applied based on the turbine location with the highest turbulence intensity. Therefore a single, high-turbulence turbine location within a project impacts the wall thickness of all towers of that project, which implies a high-cost multiplication factor.

The effective turbulence intensity was quantified using the Frandsen model, which implies that the structural load ranges vary linearly with the turbulence intensity (Frandsen, 2007). The model was used to determine the damage-equivalent bending moment[1] at the tower bottom, $M_{YT,DEL}$, as a function of the varying wind and wake conditions in the farm. The bending moment is then related to the damage-equivalent stress $\sigma_{DEL}$ and the tower wall thickness $t$ via the following equation

$$\sigma_{DEL} \propto \frac{M_{YT,DEL}}{t}. \tag{1}$$

To maintain a constant damage-equivalent stress, the wall thickness of the tower thus needs to increase proportionally to the damage-equivalent load. This can be translated to an increased wall thickness required to support the increased load

$$t_{new} = t_{old}\left(\frac{I_{eff}}{I_a}\right), \tag{2}$$

where $I_a$ is the ambient turbulence intensity and $I_{eff}$ the effective turbulence intensity that can be evaluated as

$$I_{eff}(\bar{U}_a) = \left[\int_0^{2\pi} p(\theta|\bar{U}_a)I^m(\theta|\bar{U}_a))d\theta\right]^{\frac{1}{m}}. \tag{3}$$

In this equation, $p$ is the probability of a certain wind direction occurring at hub height, $\theta$ is the wind direction, $\overline{U}_a$ is the mean wind speed at hub height, $I$ is the turbulence intensity in the wake, which consists of the ambient turbulence intensity and the wake-added turbulence intensity, and $m$ is the Wöhler exponent of the material determined by the SN-curve. The effective turbulence intensity is driven by the minimum inter-turbine spacing in the wind farm. To compare the impact of the layout, the effective turbulence intensity was calculated for the individual turbines in the regular (blue) and irregular (red) wind farm layouts using a Wöhler exponent $m = 4$. The result is shown in Figure 6.

As expected, the effective turbulence intensity levels are more constant for the regular wind farm layout. This means that wake-induced tower fatigue is similar for all turbines. For the irregular layout, however, a single outlier (in this case, turbine 67) can significantly increase tower steel for the entire project.

Conveniently, an irregular layout has a higher potential of increasing the minimum inter-turbine spacing. By strategically relocating a limited number of turbines, the wake-induced turbulence of those turbines (and thereby the tower design of all turbines) can be optimised, effectively resolving the issue. One of the turbines for each of the turbine pairs with inter-turbine spacing less than 4 RD is manually moved to satisfy a 4 RD separation constraint to show the result in wake-induced turbulence. The result is shown in Figure 7, resulting in a decrease of 10.4% of the maximum effective turbulence in the wind farm. This in turn results in a cost decrease which can go up to millions of euros depending on the steel price and number of turbines.

---

[1]The damage-equivalent load is a load with constant amplitude and fixed frequency causing the same damage the actual variation of loads over a lifetime.

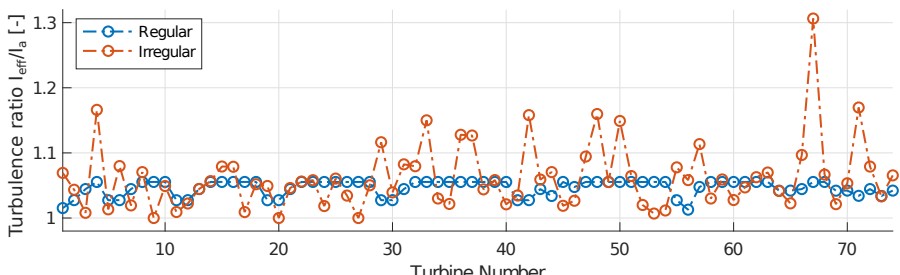

**Figure 6.** Effective to ambient turbulence intensity ratio $I_{eff}/I_a$ for the different turbines in wind farms with regular (blue) and irregular (red) layouts due to wake-added turbulence using the Frandsen model (Sickler, 2020).

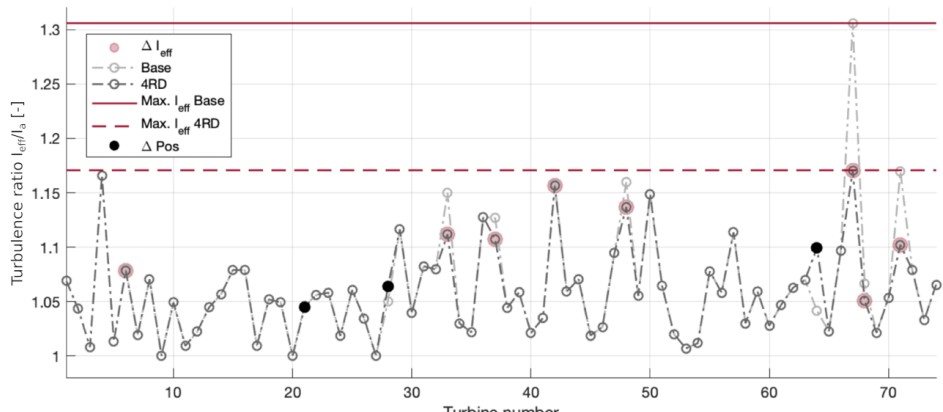

**Figure 7.** Effective to ambient turbulence intensity ratio $I_{eff}/I_a$ of the original and increased inter-turbine spacing wind farm layout. The re-positioned turbines are denoted with a filled black dot. The turbines which are not re-positioned, but do experience a change in effective turbulence, are highlighted in red. The maximum effective turbulence is indicated with the horizontal red lines for the respective layouts (Sickler, 2020).

The AEP is found to increase with + 0.043% (increasing) for the 4RD spacing compared to the 2.73RD spacing. This means both the KPI's maximum effective turbulence intensity and the AEP perform better with an increased spacing.

## 5   General applicability of results

The sensitivity analysis presented in this section serves as a method to predict the outcome of results:

– Annual energy production
– Maximum power drop
– Standard deviation power per wind direction
– Maximum effective turbulence, and
– Standard deviation of the effective turbulence

To assess the general applicability of the findings, alternative sets of regular and irregular wind farm layouts are explored. The
cases are re-analysed with a reduced rotor diameter and a uniform, unidirectional wind rose are implemented to analyse the
effect of the extreme wind rose cases. While this may not inform about the global effects of different layouts, it provides a first
step toward evaluating the general trends.

## 5.1 Rotor Diameter

To study the sensitivity to the turbine size, the rotor diameter of the IEA Wind Task 37 reference wind turbine was down-scaled
from 190.8 to 178.3 m. It is expected that the impact on relative performance for the irregular wind farm layout to the regular
layout would be small. Indeed, the difference is calculated to be a percentage point change of 0.03 % (an AEP increase of 0.66
% and 0.63 %, respectively).

For persistence to wind direction it is found the irregular wind farms perform better than their regular counterpart independent
for a smaller rotor diameter, with minor decrease of 0.3% and 0.4% for the maximum power drop and standard deviation
respectively. This is small compared to the significant change of rotor area (-13%). The results of the power output are shown
in Figure 8 below.

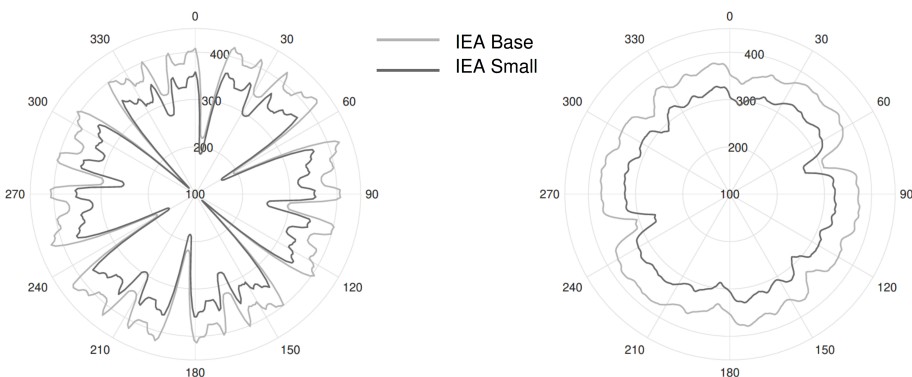

**Figure 8.** Power output in MW as a function of wind direction for the (a) Regular and (b) Irregular wind farm layout with the IEA Base
(light-grey) and IEA Small (dark-grey) turbines, with a mean wind speed of 9.5 m/s (Sickler, 2020).

For effective turbulence, a higher sensitivity to changes in the rotor diameter is found, at 5.5 % and 3.5% respectively. This
can be explained when considering that the wake model used, Frandsen, includes turbines within a radius of 10 RD around
each turbine. So, a smaller rotor diameter results in fewer neighboring turbines being considered in the calculation, resulting
in significantly lower results for effective turbulence.

Overall, assessing the rotor diameter sensitivity results it can be observed that the difference in performance of the wind
farms with irregular layout compared to the wind farms with regular layout is very similar regardless of the turbine rotor
diameter selected. However, although not changing the overall conclusion regarding irregular vs. regular layouts, turbulence
intensity is found to be impacted more significantly by the rotor size than the power performance results.

## 5.2 Wind Rose

A uniform wind rose and a single direction wind speed of 9.5 m/s in the prevailing wind direction have been used to investigate the sensitivity of the results to the climate. The percentage point change computed comparing the percentage point difference between the uniform and single direction wind speed results to the Base Case results in -0.23 % and -0.99 % for the uniform and single direction wind speed respectively.

In the analysis of persistence to wind direction, the wind farm is subjected to two different wind speeds to assess their effect on the obtained results. The wind speeds selected are 8 m/s and 12 m/s respectively. The difference in percentage point difference when comparing the irregular and regular wind farms show -0.68% and -1.85% compared to the base mean wind speed case. Also, the effective turbulence intensity is analysed for the uniform wind rose distribution. The sensitivity of the maximum effective turbulence and standard deviation of the effective turbulence are both found to be relatively high compared to the other previous sensitivity studies. With a percentage point change of 10.1 % and 9.9 %, the wind climate distribution is found to significantly change the effective turbulence results. Overall, the sensitivity of the results to changes in wind climate is significant, in particular the results regarding the effective turbulence for a (very large) change in the wind rose. Still, the irregular layout outperforms the regular one.

## 5.3 Layout

Finally, a second set of regular and irregular turbine patterns was used from the same study by Sanchez Perez Moreno (2019), and an optimisation in WindPRO was performed for the Base Case area, number of turbines, wind speed distribution and rotor diameter. These MDAO- and WindPRO-optimised farm layout pairs were compared. The aim of performing these additional case studies was to assess the global effect of these wind farm layouts on the performance indicator group results. We observed that the relative behaviour of the layout pairs (Base, MDAO, and WindPRO) is very similar.

The annual energy output difference in percentage points difference compared to the base case is found to be 0.38% and 0.01% for the WindPRO and MDAO cases respectively. Comparing the persistence to wind direction, the irregular wind farm layouts showed better performance for both the maximum power drop and standard deviation of the power output as a function of wind direction. Corresponding to a difference in percentage change of 12.4 % and 41.6 % for the maximum power drop, and 8.6% and 68.1% for the standard deviation of the power output per wind direction.

The effective turbulence intensity comparative analysis for the MDAO and WindPRO layouts show a percentage point change of 5.9 % and 1.7%. Overall, the effective turbulence intensity results for the additional wind farm layouts indicate that the trend found in Chapter 4 applies for a different set of regular vs. irregular layouts as well.

The global sensitivity study performed with the additional layout case studies is shown on the right of this line. The changes to the conditions and layouts are explained in the discussion of the results below.

The conclusions drawn from this sensitivity study appear to be more generally applicable for irregular patterns. It is found that for different layout cases, wind roses and rotor radius, the absolute inter-pair results show the irregular layout outperform-

ing the regular one except for maximum effective turbulence, which is worse for irregular layouts and which can be explained very well.

## 6   Conclusions

Based on the work done in this paper, it is found that irregular wind farm layouts outperform regular layouts regarding energy production, as overall wake losses are reduced. In the performed case study, an overall yield increase of 0.66% was found for the chosen layout pair. A notable finding is that irregular layouts also increase the persistence to wind direction, which means that the power output is less sensitive to fluctuations in wind direction. For the case study done, the maximum power drop of the irregular layout is roughly one-third of the maximum power drop observed in the regular layout. This characteristic

of irregular layouts improves the predictability of the power output, reducing the impact of wind forecasting errors on power system operation and potentially increasing the value of power in the electricity market. The actual benefit of an irregular layout compared to a regular one, ceteris paribus, will vary between the actual layouts compared.

A drawback of irregular layouts is that the turbulence intensity for some turbines may be higher. Using the Frandsen model, the irregular layout is found to generate 14 to 24% higher worst-case wake-induced turbulence levels. This results in higher

fatigue loads, increases in tower wall thickness and therefore higher steel costs. Since typically only one wind turbine tower design is used per project, this effect may be significant at the project level. By increasing the minimum spacing, the worst-case turbulence intensity is reduced by 10.4%, bringing the worst-case wake-induced turbulence within the range of the regular layout. This conclusion has general validity since irregular layouts inherently have at least a few turbines positioned with limited spacing. Although the use of irregular wind farm patterns increases energy yield, improving the performance of the

wind farm as a whole requires caution to inter-turbine spacing to limit the negative effect of increased fatigue-loading.

*Author contributions.*   MS and BU wrote the original draft. MZ, RS and KD were involved in the revision of the work.

*Competing interests.*   There are no competing interests.

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
