# Peer review of "Offshore wind farm optimisation: a comparison of performance between regular and irregular wind turbine layouts"

_Wind Energy Science, 2022_

## Author Response (AR1)

**Response to reviewer #1, Andrew P.J. Stanley:**

*First of all, thank you for the clear and complete set of review comments, which were very constructive. We have incorporated all of them in the paper text. We will be using Italic font to reply to your comments.*

The paper compares several performance metrics of a regular and irregular wind farm layout. The authors conclude that irregular layouts are better in terms of energy production and power variability as a function of wind direction, but regular layouts are better in terms of fatigue on the turbine towers.

**My Main Conclusion**

While the paper has some interesting information, I don't think any general conclusions about regular versus irregular layouts can be drawn from what has been presented.

First, because most of the paper introduces only two turbine layouts for a single location, Section 5 should be the most important section that presents and defends why any of the observations are globally applicable. However, Section 5 section does not convince me. Part of this could be that I don't fully understand what was done, I had to read through the section several times and am still unsure if I understand correctly. How many additional cases were considered? What are the light and dark gray bars in Figure 8? I think for anyone to agree that the observations previously made between regular and irregular layouts, there need to be several demonstrations for a wide variety of wind plants (size, boundary shape, several wind roses, objective…).

*We have rewritten Section 5 to address the above and to clarify which aspects are considered to be generally applicable and which may not be. Also we have included an explanation of the three scenarios: Rotor Diameter, Windrose, Different Case Layouts. The text in Figure 8 now explains per sub-figure what is shown.*

Second, the majority of the paper compares one regular and one irregular layout, which were taken from another paper, and I assume, were optimized for some objective.

*Your assumption is correct.*

With this information alone, I wouldn't conclude that the observations made are general to all regular and irregular layouts, even for this specific location. What if I optimized a grid for a different objective, or included some additional constraints? Same with the irregular layout? What if I used a different regular layout that wasn't a grid? The observations of this paper may be general, but I don't think that should be concluded without more evidence.

*In the sensitivity study, further explanations are provided on which aspects are, and which aspects may be, generally applicable. Additional layouts have been analyzed, which show similar results, indicating general applicability. Indeed, this also depends on the degree of irregularity of the layout, and this nuance has been added, too.*

***A Few Other Things***

- Line 45 – "Third, an optimisation of the farm layout inherently leads to an irregular pattern, as shown by most optimization studies in the literature." This is not necessarily true. It depends how you setup the problem. Plenty of wind farms are optimized with a regular grid.

  *Changed to 'Third, an optimisation of an existing regular wind farm pattern inherently leads to an increase in irregularity, as shown by optimisation studies'*

- Figure 5 and the related discussion – Is this the right metric? This gives no indication of the probability of a high-power fluctuation. I agree that a gridded layout will have some wind directions that are associated with very high losses, but if the grid is optimized those directions will be associated with very low probability. I'm not convinced that "The difference between the imbalance cost of the regular and irregular wind farm would then become visible within this 1.6 %." Small note, I would put the two subfigures on the same plot, it would be easier to see the differences and conserve some space!

  *Thank you for the suggestion; the figure was changed into one figure including both regular and irregular. It is noted that a regular grid, even if it is optimized for specific wind directions to minimize this effect, will still be prone to larger power fluctuations. Generally, wind climate data is not analyzed chronologically to capture this effect. The effect manifests itself close to the average wind speed, so the probability of occurrence is likely still significant, and the impact can also be significant.*

- Line 109 - "Interestingly, while monopile foundation designs are optimised for individual locations within an offshore wind farm, typically only a single tower design is applied based on the turbine location with the highest turbulence intensity." This seems like an opportunity to optimize tower designs for single locations, rather than save enormous amounts of money by reducing the turbulence of the worst-case tower location. It seems silly to hugely overdesign every tower because of one very poor instance.

  *Based on the author's experience in wind park engineering, the opportunity for tower design optimization is limited to none. Typically, integrated support structure design (tower + monopile, either with or without transition piece) is a very time-, resource- and supply chain-constrained process. Primary steel needs to be certified at the financial close of a project, and design iterations typically take 1-1.5 years. The tower's primary steel design also defines the section flanges and the tower internals, which require engineering, procurement, and manufacturing processes as well. Finally, having one standard tower is very valuable for streamlining installation logistics. To achieve the required economies of scale, standardization is key. For these reasons (which are not scientific, but practical) the offshore OEMs will all deliver one tower per offshore project (and will try to re-use an earlier tower design when possible). Considering the highest turbulence location in the permitting/siting phase of the project is a quick win.*

- Already mentioned above, but I have very little idea what Figure 8 means.

  *We adjusted the text with a figure.*

- Line 168 – "Irregular wind farm layouts outperform regular layouts regarding energy production…" this information is still worth including, but it is important to remember that this is true by definition. Less constrained problems either perform just as well or better than more constrained problems.

  *Agreed. Reference to earlier sentence.*

- Line 169 – "A notable finding was that the irregular layout also increases the persistence to wind direction…" again already mentioned above, but is this still notable when the directional probabilities are considered?

  *Agree, added nuance to windrose probability distribution.*

- Figure 4 and 7 are really interesting!

  *I recognize that I may not have understood everything as was intended, and I may not have communicated my thoughts as clearly as I would have liked. If anything is unclear or you disagree with anything I have said, please reach out to me and we can continue this discussion.*

- Figure 4 and 7 are really interesting!

I recognize that I may not have understood everything as was intended, and I may not have communicated my thoughts as clearly as I would have liked. If anything is unclear or you disagree with anything I have said, please reach out to me and we can continue this discussion.

PJ Stanley

*Noted. We trust that these clarifications and the revisions of the texts and figures capture your comments, but we are open to discuss further.*

**Response to reviewer #2, Christopher Bay:**

This manuscript presents some analyses of performance of regular and irregular wind farms, proposing that irregular wind farms outperform regular wind farms in several performance indicators. The overall topic is of interest to the wind energy community, but I believe the manuscript needs to be improved significantly to be considered for publication. The analyses, while interesting, need to be broadened in scope to better draw general conclusions. I have provided specific comments below.

*Thank you very much for your in-depth analysis and read-through of this paper. This is very much appreciated. Unfortunately, there was only limited time available for this work because graduation projects are limited to 8 months. Some of the substantiation and further research, therefore, was not performed within this available time window.*

1. I agree with the authors that the irregular wind farm studied in Section 3 outperforms the regular wind farm for energy yield and predictability, but I do not think the authors included sufficient discussion to justify the irregular farm outperforms in value to the electricity market. More discussion should be included to illustrate to the reader how this is true.

   *Correct that the quantification of the value to the electricity market is not presented in this study. This was not part of the research objective and thus was not included more elaborately. This highly depends on the electricity market and the amount of renewable energy connected to the grid.*

2. Section 4 would be strengthened by including discussion around the changes in performance of the irregular farm after the turbines have been shifted by the 4 RD. Most likely, the change in energy production is minimal, but would be good to show.

   *This quantification is indeed of added benefit to the research; thanks for pointing this out. Chapter 7 of the MSc thesis elaborates extensively on the 4RD spacing, the increase in effective wake-added turbulence, and the minor change of AEP. The difference in AEP is +0.043% (increasing) for the 4RD spacing compared to the 2.73RD spacing. The effective turbulence decreases with -10.4% compared to the base case (2.73RD spacing), meaning both the AEP and the maximum effective turbulence intensity increase as KPI. This is now added to Section 4 of the paper.*

3. Section 4 would also be strengthened by including an approximation in the difference in tower cost between the 3 layouts (regular, irregular, and repositioned irregular). Discussion on the change of the repositioned irregular layout compared to the regular layout would be beneficial as well.

   *This is highly dependent on the steel price, as this has significantly fluctuated over the past year I would like to steer clear of quantification in terms of cost. However, a high-level indication can be given, which I have added to Section 4.*

4. Figure 8 needs a legend to indicate what the light gray and darker gray bars indicate. I also wonder if this information would not be more readable in a table. With the different y-axis

scales between the plots, it is difficult to compare relative impacts across parameters. Also, some changes are so small they do no appear on the plots.

*Added the legend to the figure. Using a different y-axis would not show the full range of the power drop.*

5.  Section 5 would be significantly improved with more details and/or a better illustration of the different cases being examined. As written, it is difficult to follow. Suggestions include: 1) including a figure that shows the two wind roses used in the analysis so the reader can understand the difference between the uniform and unidirectional wind roses, 2) a diagram or flow chart depicting the changes made between the cases; this is currently only in the text and is difficult to follow, and 3) a more clear description of the analysis occurring with the MDAO- and WindPRO-optimized layouts; the authors state that the farm layout pairs were compared, but I am unsure of what the pairs are and how they vary across Base, MDAO, and WindPRO.

    *Another reviewer made the same comment. We have made several adjustments to Section 5 to improve the wording.*

6.  The conclusions drawn hold for the two cases examined in the manuscript, but are not proven for all cases, and should be noted as such. An interesting and worthwhile addition to this study would be an analysis of a larger sampling of irregular wind farm layouts which would help prove or disprove the conclusions in a more general sense. Additional analysis across other wind roses would also be helpful.

    *Added as an additional sentence.*

7.  A discussion of other potential effects of an irregular wind farm layout should be included as well, or at least acknowledged that an irregular layout can have other effects such as changes in cabling costs and impacts on navigability of ships through a wind farm. These two topics are not exhaustive of all the effects of irregular layouts compared to regular layouts.

    *Agreed. In the original graduation project, I did look into the effect on the inter-array cable layout but found that the changes are marginal compared to the other KPI's. Overall, the inter-array cable layout can be adjusted well to accommodate different layouts without significant impacts on overall length. Other aspects, such as local soil conditions, cable/pipeline crossings, and archaeological findings that must be avoided, can have equal or larger impacts on cable length. Therefore, we have decided (also for word count) not to include this in the paper. The irregular grid has a total increased length of 2.19% compared to the regular positioning of the turbines. Additionally, offshore wind turbines are often spaced at such distances that the navigability of the wind farm does not improve with a regular or irregular turbine layout. The pattern with 'too' large distances is not recognizable for ships or others.*

**Other comments/corrections:**

- Line 58: Are there four levels in Figure 2? Or three levels? Are there two levels within the PI group? If Electricity Price is on a different level than Annual Energy Production, it is difficult to discern that through the flow of the arrows.

  *Changed the wording to three levels to avoid confusion. Three levels, with sub-categories from level three. The arrow splits before reaching annual energy production. There is not sufficient space to place this level all on the same line, unfortunately.*

- Line 61: Consider rewording; as it reads, the first item does not fit grammatically.

  *Changed the sentence into two separate sentences to make it grammatically correct.*

- Line 62: Are there five sub-performance indicators? I see seven in Figure 2.

  *There are seven Performance Indicators (PIs). The bullets below are sub-performance indicators, of which five have been selected. Those five have been grouped to represent 'Power Performance' and 'Wake-Induced Tower Fatigue'. The wording has been amended to reflect that more clearly.*

- Line 100: Typo in "indicator".

  *Changed.*

- Vectorized images are preferred for readability.

  *We will align with the publisher to ensure good readability.*